# Experimental investigation of shape-enhanced rotating cylinders with electric heaters and solar panels for augmented pyramid solar still performance

A. S. Abdullah[1], Mutabe Aljaghtham[1], Wissam H. Alawee[2]*, Z. M. Omara [3]*, Fadl A. Essa[3]

1 Department of Mechanical Engineering, College of Engineering in Al-Kharj, Prince Sattam bin Abdulaziz University, Al-Kharj, Saudi Arabia, 2 Control and Systems Engineering Department, University of Technology, Baghdad, Iraq, 3 Mechanical Engineering Department, Faculty of Engineering, Kafrelsheikh University, Kafrelsheikh, Egypt

* zm_omara@yahoo.com (ZMO); wissam.h.alawee@uotechnology.edu.iq (WHA)

## Abstract

This work investigated the influence of various amendments on a pyramid solar still with rotating cylinders (RCPSS) and rotating corrugated cylinders (RCCPSS). We compared distillate yield from the RCPSS with a baseline design (PSS) to assess the effectiveness of each modification. The study explored incorporating reflectors, silver nanoparticle-infused phase change material (PCM-Ag) composites within the cylinders, and a vapor-withdrawing fan with an external condenser. In addition, three electric heaters were fixed on the basin water to raise its temperature. The energy required to run the heaters was captured from a PV system. Also, the effect of covering corrugated cylinders with wick on the performance of the modified still was also studied. The RCCPSS significantly outperformed the PSS, producing 8500 mL/m² of freshwater daily compared to 3100 mL/m², representing a 174% increase in distillate. Additionally, heaters and PCM-Ag composites further enhanced distillate yield by 244% and 365%, respectively. However, the most optimal configuration involved combining a wick, heaters and fan. This setup yielded the highest distillate production (14950 mL/m²), a 382% increase over PSS, and achieved a thermal efficiency of 75%. Finally, the freshwater production cost was lower for the RCPSS with wick, heaters and fan ($0.01/L) compared to the PSS ($0.02/L). The work demonstrates a strong commitment to advancing the United Nations Sustainable Development Goals (SDGs), particularly SDG 6: Clean Water and Sanitation.

## 1 Introduction

Growing populations, rapid industrialization, and escalating energy demands are exerting unprecedented pressure on the planet's finite freshwater reserves [1,2]. Studies project a

**Data availability statement:** All relevant data are within the paper.

**Funding:** The author(s) received no specific funding for this work.

**Competing interests:** Yes.

dramatic surge in global water consumption by 2050, intensifying challenges for billions already lacking access to safe drinking water [3,4]. In response, desalination has emerged as a critical alternative to bridge this resource gap [5,6]. Desalination involves the removal of dissolved salts from saline water sources—primarily seawater or brackish supplies—to yield potable water or industrial-grade freshwater [7–10]. Recognized as a pivotal tool in addressing water scarcity, this technology is gaining traction worldwide [11,12]. Notably, solar-driven desalination has risen to prominence as a sustainable solution, aligning with global initiatives to adopt renewable energy and promote environmentally responsible water management [13–15]. Solar desalination systems fall into two broad categories [16–18]: (1) Direct methods: Solar energy directly heats saline water, inducing evaporation and subsequent condensation to produce freshwater [19,20]. (2) Indirect methods: Solar collectors concentrate thermal energy to power a separate desalination process (e.g., multi-stage flash or reverse osmosis) [21–23]. While both approaches rely on evaporation-condensation principles [24–26], indirect methods have garnered significant attention due to their economic viability and scalability [27–30].

Single-basin SSs offer several advantages, including low cost and simplicity [31]. Nevertheless, the distillate is generally restrained to around 3 liters [32–35]. Researchers have explored diverse pattern changes that focus on enlarging the surface exposed for evaporation [36–41]. These innovations incorporate a range of geometrical characteristics, like discing [42], trays [43], dishing [44,45], vertical [46,47], stepped [48–51], wicking [52], cylinder [53,54], tubular [55,56], semi-spherical [57–59], & pyramid [60–63] designs.

Recent advancements in solar distillation technology have moved beyond conventional surface-area expansion to incorporate sophisticated, multi-faceted engineering approaches for enhanced performance. Contemporary research has identified four major innovations driving this progress: geometrically optimized basin liners that improve interfacial thermal dynamics [64,65]; nanostructured surface coatings designed to maximize broadband solar absorption while minimizing radiative losses [66]; strategically positioned reflective concentrators that intensify incident solar flux [67–69]; and phase-change material (PCM) composites that provide thermal energy buffering during variable irradiation conditions [55,70]. These system-level modifications collectively establish an integrated energy management framework where enhanced photon capture, reduced parasitic heat dissipation, and dynamic thermal regulation work synergistically to improve both peak productivity and operational stability under fluctuating insolation. Additional performance gains have been achieved through the incorporation of advanced nanomaterials for improved light absorption [71,72], rotational components for optimized thermal distribution [40,73], and innovative condensation structures to maximize vapor-to-liquid conversion [74,75]. Together, these technological developments have led to substantial improvements in distillate output [76], demonstrating the significant potential of next-generation solar desalination systems to address global water scarcity challenges. The integration of these innovations represents a paradigm shift in solar distillation, combining multiple optimization strategies to achieve unprecedented efficiency and reliability in freshwater production.

The absorber architecture has emerged as a pivotal factor governing solar still (SS) performance, with contemporary studies substantiating its significant impact on interfacial energy transfer dynamics [77]. Recent comparative investigations of advanced absorber configurations, including corrugated surfaces, capillary-driven wicking structures, and phase-change material composites, have revealed substantial performance enhancements [78]. Notably, these designs demonstrated a remarkable 154% improvement in distillation efficiency, highlighting the nonlinear performance scaling achievable through optimized surface-thermodynamic interactions. Complementary research on geometric optimization has shown equally promising results, with a pyramidal distiller incorporating internal fin arrays achieving an 82.1% productivity increase compared to conventional designs [79], underscoring the critical role of turbulent flow modulation in enhancing evaporative mass transfer. Further breakthroughs were achieved by Hemmatian et al. [80], who integrated gravity-assisted heat pipe arrays into SS systems, yielding a 40.7% boost in absolute water production (2248 ml/m²) accompanied by a 20.3% improvement in overall system efficiency. These collective advancements demonstrate the effectiveness of multifaceted engineering strategies – encompassing surface functionalization, structural geometry optimization, and thermal bridging techniques – in establishing new performance benchmarks for solar desalination technologies. The synergistic integration of these diverse approaches presents a transformative pathway for overcoming the efficiency limitations inherent in conventional solar still designs.

Material engineering has emerged as a pivotal strategy for augmenting solar still performance through advanced surface and structural modifications. Shanmugan et al. [81] systematically evaluated evaporative surface treatments, including $TiO_2$ nanoparticle coatings, chromium(III) oxide ($Cr_2O_3$) deposition, and hybrid adsorbent integration. Their optimized configuration demonstrated a seasonal variation in freshwater yield, with daily production ranging from 5390 mL during winter to 7890 mL in summer under Mediterranean climatic conditions. Concurrently, researchers have extended performance optimization beyond material composition to innovative system architectures. Essa et al. [82] developed a cascaded distillation chamber design, achieving a 50% improvement in thermodynamic efficiency and 114% increase in daily productivity compared to conventional single-stage units. Further advancements include the implementation of sub-atmospheric pressure regimes within evaporation chambers, with experimental trials demonstrating a 63% enhancement in distillate output through vacuum-assisted vapor extraction [83]. The feasibility of low-cost fabrication by utilizing locally sourced materials was validated [84], attaining a sustainable yield of 1700 mL/day while maintaining operational durability. In parallel, the introduction of functionalized working fluids has shown transformative potential: antibacterial magnetized nanofluids exhibited a 218% productivity surge under standardized irradiation tests [85], while $Al_2O_3$ nanoparticle suspensions in heat transfer fluids boosted yield by 12.2% relative to non-nano-enhanced analogs [86].

Emerging kinematic modifications to solar still architectures have demonstrated substantial gains in interfacial energy harvesting. A progressive innovation involves replacing static basin liners with axially rotational disc arrays [87,88], where continuous surface renewal mitigates thermal boundary layer stagnation. Experimental validations report a 124% enhancement in freshwater yield relative to stationary designs, attributed to augmented evaporative surface area and convective mixing. Parallel investigations into modular tray systems [89–93] reveal analogous benefits, with vertically staggered trays achieving a 105% productivity increase over conventional configurations by optimizing vapor diffusion pathways [91]. Further performance leaps have been achieved through dynamic component integration. Rotational spherical bodies within evaporation chambers [94,95] induce turbulent flow regimes, disrupting thermal stratification and amplifying latent heat recovery—yielding a 103% output improvement under controlled irradiance [96,97]. The most pronounced advancement combines geometric optimization with hybrid energy storage: pyramidal liners coupled with specular reflectors and PCM composites in PSS achieved a 142% distillate surge [98], demonstrating the synergistic potential of optical concentration, thermal inertia modulation, and latent heat recycling.

This study introduces a novel approach to enhancing pyramid solar still (PSS) performance by incorporating rotating cylinders—specifically, rotating plain cylinders (RCPSS) and rotating corrugated cylinders (RCCYPSS)—a concept not previously explored in the literature. Unlike conventional PSS systems, this work uniquely investigates the synergistic

effects of rotational motion, surface corrugation, and advanced modifications such as PCM-Ag nanocomposites inside the cylinders, a vapor extraction fan with an external condenser, and reflector integration. A key finding is the identification of distinct optimal rotational speeds: 0.5 rpm for plain and corrugated cylinders without wicks [73], and 0.05 rpm for corrugated cylinders with wicks [67], irrespective of heater usage—a critical insight that has not been reported before. Furthermore, the combined use of rotational dynamics, enhanced heat transfer via PCM-Ag, and vapor extraction mechanisms presents a significant advancement over traditional passive solar still designs, offering a new pathway for efficiency improvement in desalination technologies.

## 2  Materials and methods

### 2.1  Setup assembly

The experimental setup, detailed in Figs 1 and 2, employed two identical solar stills. One served as the conventional pyramid distiller, while the other was the modified distiller with rotating cylinders (RCPSS) and rotating corrugated cylinders

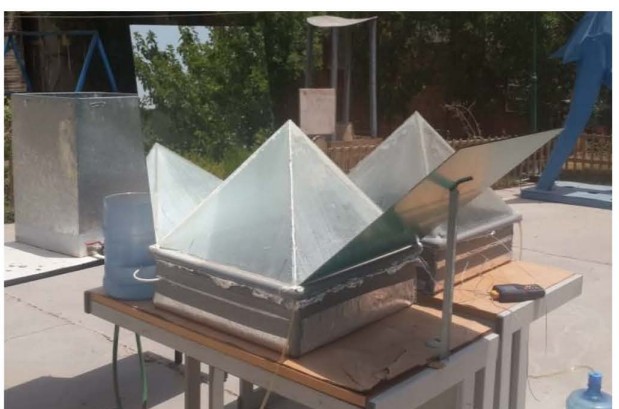

Tested distillers with mirrors Corrugated cylinders

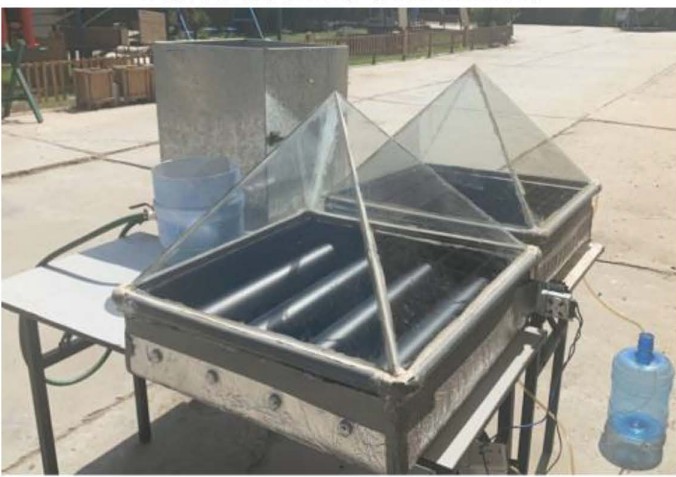

Tested distillers Tested distillers with PV cell

**Fig 1.  Images of experimenting layout.**

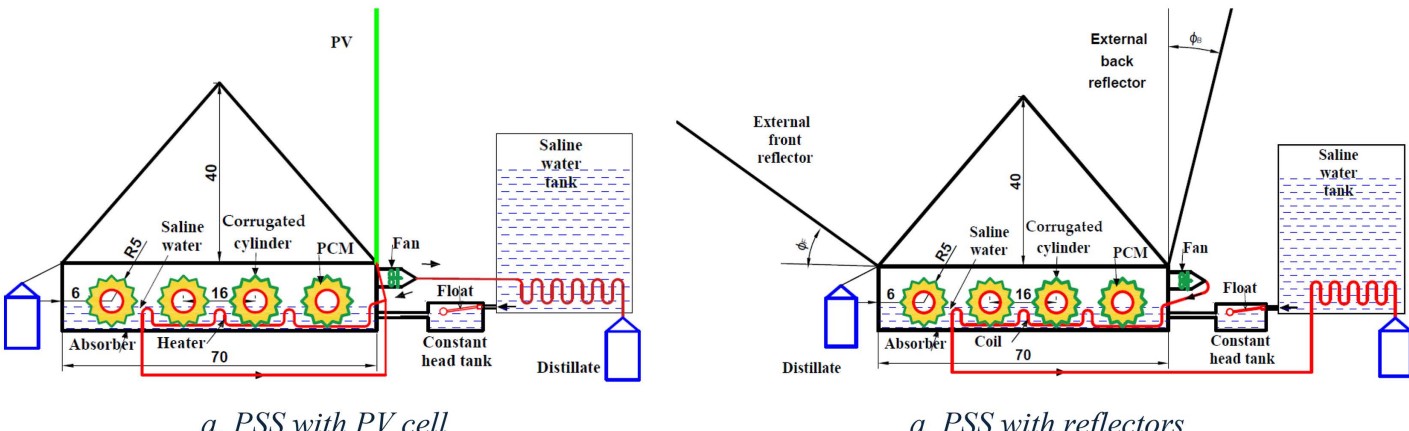

*a. PSS with PV cell*          *a. PSS with reflectors*

**Fig 2. 2D plot of RCCPSS.**

(RCCYPSS). Additionally, the setup included a feed tank, a fixed level tank to attain water level within the distillers constant, and necessary measuring instruments. A general designing parameters' discerption is included in Table 1.

The experiment utilized two solar stills constructed from 1.5 mm steel sheets. Both stills shared a square base with identical areas of 70×70 cm and vertical walls measuring 15 cm in height. To create the pyramid shape, each still featured a square basin topped with four triangular glass covers welded together (Fig 1). The 40 cm tall glass stood at a 49° tilt angle from the horizontal base. This angle is optimal for two reasons: it maximizes surface area of glass for solar exposure, and it facilitates the swift movement of condensed water droplets down the glass towards the bottom, promoting efficient condensation [68]. Four troughing parts are positioned at triangles' bottoms to gain distilled drops, which were then channeled outside the stills and accumulated in graduated flasks. To minimize heat loss, the external basin surfaces were insulated with 5 cm thick fiberglass.

Unlike the modifications discussed previously, the RCPSS maintained the same base area and dimensions as the reference PSS. The investigation explored the RCPSS performance under three distinct operating conditions: incorporating reflectors, integrating a fan with a condenser, and utilizing PCM infused with Ag-nanoparticles within the rotating cylinders. The PCM utilized was the commercial paraffin wax with a quantity of 9.6 kg. Notably, the defining feature of the RCPSS remained the inclusion of four internal rotating cylinders. The rotating cylinders, a key modification in the RCPSS, were crafted from 0.5 mm thick aluminum sheets. Each cylinder measured 10 cm in diameter and 68 cm in length. A chain system connected the four cylinders, allowing them to rotate simultaneously. To power the DC motor and achieve the desired cylinder rotation speed, a photovoltaic (PV) system was employed. This system utilized polycrystalline silicon PV panels to generate electricity. A speed controller ensured the pulley maintained required rotational velocity. This highlights the potential for a more economical desalination system by replacing the PV system with a smaller unit (around 10 W) when solely powering the motor. Table 2 displays the characteristics of the wax and the wax containing 2.5 weight percent nanoparticles.

In addition, the external condenser in the RCPSS setup utilized a 6-watt axial fan to take out extra vapor from basin water area (Fig 2). This axial fan measured 10 cm in diameter (total width) and had 9 cm diameter blades. The extracted vapor then traveled through a copper coil submerged in the basin water, preheating it before exiting. Finally, the vapor passed through the feed water tank where it condensed and collected in a graduated bottle. In essence, the external condenser functioned as a feed water tank with a copper coil. Hot vapor passed through the coil, transferring heat to the cooler feed water acting as a coolant. The condensing unit consisted of a 3.5-meter long, 0.025-meter diameter copper tube coiled within the tank. The coil length was determined based on the heat transfer capacity needed to condense the

**Table 1. Design dimensions of PSS and RCCPSS.**

| # | Parameter | Value |
|---|---|---|
| 1. | Feeding reservoir | 50×50×100 cm |
| 2. | Body structure | 1.5 mm steel |
| 3. | Base heights – PSS and RCCPSS | 15 cm |
| 4. | Base dimensions – PSS and RCCPSS | 70×70 cm |
| 5. | Land area – PSS and RCCPSS | 0.5 m² |
| 6. | Glazing tilting – PSS and RCCPSS | 49° |
| 7. | Isolation – PSS and RCCPSS | Glass wool |
| 8. | Mirrors dimensions | 70 cm×70 cm |
| 9. | Condensing copper tube coiled | 3.5 m long, 0.025 m diameter |
| 10. | Cylinders structure | 0.5 mm thick aluminum sheets. |
| 11. | Cylinder dimensions | 10 cm in diameter and 68 cm in length. |
| 12. | Wick | Jute wick material |

**Table 2. Paraffin wax's thermophysical characteristics with and without silver nanoparticles.**

| Property | PCM with Ag-nanomaterials | Pure PCM |
|---|---|---|
| Density, kg/m³ | 962 | 876 |
| Melting, ºC | 53 | 54.5 |
| Latent heat, kJ/kg | 182 | 190 |
| Specific heat, kJ/kg °C | 2.01 | 2.1 |
| Conductivity, W/m °C | 0.3 | 0.21 |

vapor through free convection using the measured temperatures of the cooling water and incoming vapor. It's important to note that this design modification, particularly the coil length, may influence water distribution and absorption within the distiller. This in turn could affect the evaporation and condensation processes, ultimately impacting the productivity.

Besides, to maximize solar rays capture, the RCPSS setup incorporated two adjustable external reflectors strategically positioned for optimal sun exposure (Fig 2). One mirror, referred to as the "external back" reflector, was placed behind the still, while the other, the "external front" mirror, was positioned in front. Both mirrors measured 70 cm×70 cm. Drawing on previous research [99,100], the mirror tilt angles were adjusted throughout the year to align with the changing sun angles across season. This dynamic approach ensured maximum solar radiation capture during winter (slightly forward tilt) and summer (slightly backward tilt). The research also emphasized the importance of elongating and slightly tilting upwards the front flat mirrors for year-round efficiency. This meticulous adjustment of mirror angles reflects the focus on maximizing solar energy capture and achieving peak system performance.

Moreover, a photovoltaic system was used to provide the DC motor by the required electric power to rotate the cylinders at the target rotational speed, Fig 1. The type of the used PV system was Poly-crystalline silicon PV panels. Furthermore, a speed controller was utilized to set the driver pulley at the wanted target speed. Additionally, as illustrated in Fig 2, an electric heater was fixed in the middle of basin water below the rotating cylinders to raise the basin water temperature. The electric heater was operated by the energy coming from the PV system. The PV panel can provide a power of 120 W. While we used three electric heaters with a power 30 W for each. In addition, the electric motor consumes 6 W only. To power the DC motor and achieve the desired cylinder rotation speed, the photovoltaic (PV) system was employed. This system utilized polycrystalline silicon PV panels to generate electricity. A speed controller ensured the pulley maintained required rotational velocity. This demonstrates the potential for developing a more cost-effective desalination system by replacing a conventional photovoltaic (PV) setup with a smaller unit, approximately 15 W, sufficient to power only the

motor. In this study, the DC motor consumes 6 W. A photovoltaic solar panel was used to independently operate both the motor (6 W) and an extractor fan (6 W) with the 3 electrical heaters, eliminating the need for any external or conventional energy sources.

## 2.2 Measurement tools

Solar irradiation is meticulously measured by pyranometer (accuracy: ± 1 W/m², range: 0–5000 W/m²). Thermocouples are utilized to know temperatures (±0.5°C accuracy, range: −50–160°C). Additionally, air velocity is known by a precise anemometer (accuracy: ± 0.1 m/s, range: 0.1 to 30 m/s). The distillate liquid is utilized with graduated bottles (accuracy: ± 1 mL, range: 0–2000 mL).

## 2.3 Uncertainty analysis

Uncertainty analysis is a structured method used to assess and quantify the uncertainties present in experimental measurements and calculated results. This approach involves mathematically propagating uncertainties from individual measured variables to the final derived quantity. For example, if a parameter Q is a function of multiple variables $(x_1, x_2, \ldots, x_n)$, expressed as $Q = f(x_1, x_2, \ldots, x_n)$, the total uncertainty in Q (denoted as ΔQ) can be estimated using uncertainty propagation laws. When the variables are independent, the combined uncertainty is calculated as follows [101,102]:

$$\Delta Q = \sqrt{\left(\frac{\partial f}{\partial x_1}\Delta x_1\right)^2 + \left(\frac{\partial f}{\partial x_1}\Delta x_2\right)^2 + \ldots + \left(\frac{\partial f}{\partial x_1}\Delta x_n\right)^2}$$

In this context, $\frac{\partial f}{\partial x_i}$ represents the partial derivative of the function $f$ with respect to each variable $x_i$, while $\Delta x_i$ denotes the uncertainty associated with $x_i$. By applying these principles, uncertainties in experimental measurements—such as solar irradiance, temperature, airflow, and distillate output—can be propagated to determine the overall uncertainty in derived parameters like thermal efficiency and productivity of a solar still. This approach ensures that results are presented with well-defined confidence intervals, improving the reliability and reproducibility of experimental conclusions. Consequently, the system's efficiency was computed using the appropriate equations, yielding an estimated uncertainty of ±3.5% in the final results.

## 3 Results and discussion

### 3.1 Operation and assessment of rotating cylinders PSS (RCPSS)

Solar intensity reached a peak of 1200 W/m² at 13:00. Fig 3 illustrates the variations in weather and operating conditions for PSS and RCPSS operating at 0.5 rpm. The PSS water temperature is consistently more than RCPSS by 0–7.5°C, reaching 62.5°C at 14:00 compared to 55°C in the RCPSS. Conversely, the RCPSS exhibited a 0–4.5°C increase in glass temperature compared to the PSS, with values of 47°C and 42.5°C, severally, at 14:00. This difference is attributed to the higher vapor generation rate within the RCPSS. The ambient air speed ranged from a minimum of 0.15 m/s to a maximum of 4.5 m/s, with an average daily speed of 1.85 m/s.

The RCPSS held less water compared to the PSS due to the volume displaced by the submerged cylinders. Despite having a lower water volume, the RCPSS exhibited lower water temperatures than the PSS. This phenomenon can be attributed to the cylinders absorbing a significant portion of the incident solar energy. This absorbed energy likely undergoes a phase change, acting as latent heat to vaporize water layer on cylinder surfaces. So, the cylinders do not contribute significantly to raising the bulk water temperature because of the evaporative process. Furthermore, the reduced water surface area exposed to solar radiation in the RCPSS (32.5 cm × 70 cm²) compared to the PSS (70 cm × 70 cm²)

 

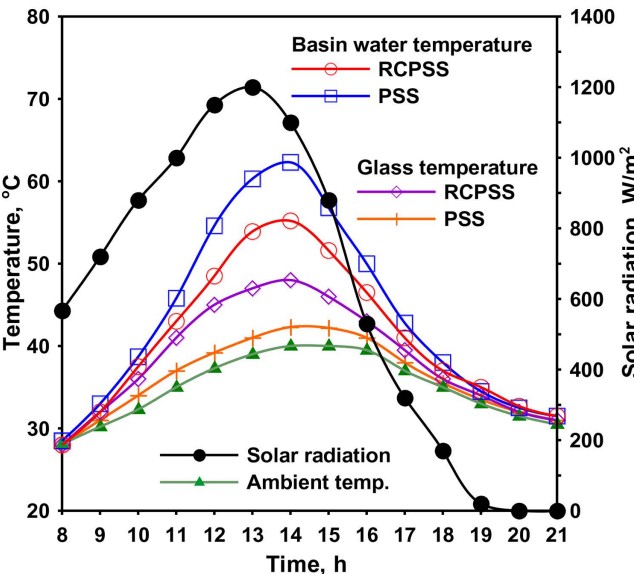

**Fig 3. Weather and operating conditions for PSS and RCPSS.**

contributes to its lower water temperature. Additionally, the shadow cast by cylinders on RCPSS water further reduces its water temperature relative to the PSS. Conversely, the higher evaporation and condensation rates within the RCPSS likely elevate its glass temperature.

Fig 4 depicts the distillate variations for PSS & RCPSS at a rotation speed of cylinder of 0.5 rpm. Interestingly, the RCPSS exhibited higher freshwater productivity compared to the PSS, despite having a lower water temperature. This can be attributed to two key factors. First of all, the RCPSS possesses a larger overall evaporation surface area compared to the PSS. When disregarding the shadow effect, the total evaporative area in RCPSS is 32.5x70 + 95x68 = 8735 cm², exceeding the PSS's area of 70 × 70 = 4900 cm². It's important to note that the values 95 (23.75x4) cm × 68 cm² represent the evaporate surface areas represent the evaporates surface areas the part not immersed in water of rotating cylinders and 32.5 (6.5x5) cm × 70 cm² represent the evaporates surface areas water, severally. The incorporation of the rotating cylinders increased the vaporization surface area by approximately 78.3%, leading to enhanced evaporation within modified PSS. As is well-known, increased evaporation within a still correlates with higher distillate productivity.

To maximize solar heat absorption, the cylinders were coated in black paint. During rotation, thin water films formed on their circumferences and rapidly vaporized because of elevated cylinder temperatures. As expected, water production peaked at 14:00, with the RCPSS and PSS achieving maximum values of 1400 mL/m² & 750 mL/m², severally. Fig 4, further confirms that the total daily freshwater production of the RCPSS (7450 mL/m²) surpassed that of the PSS (3050 mL/m²). It means a 144% augment in distillate for RCPSS. This significant enhancement is primarily assigned to the superior vaporization rate accomplished by RCPSS compared to PSS.

### 3.2 Operation & assessment of rotating corrugated cylinders PSS (RCCPSS)

The incorporation of the corrugated cylinders increased the vaporization surface area by approximately 162%. Where, the total evaporative area in RCCPSS is 12850 cm², exceeding the PSS's area of 4900 cm². As is well-known, increased evaporation within a still correlates with higher distillate productivity. The engineered surface morphology of corrugated cylinders enhances solar energy harvesting through two synergistic mechanisms: (i) geometric amplification of effective

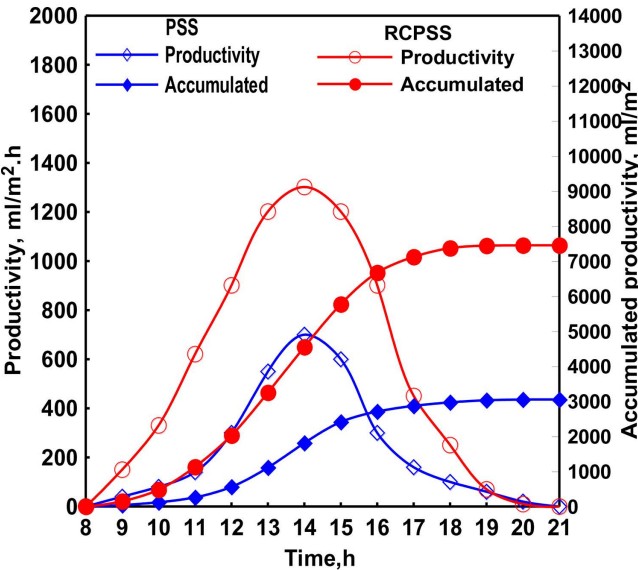

**Fig 4. Changes of distillate of PSS & RCPSS.**

irradiance exposure via modulated surface topology, and (ii) enhanced photon capture efficiency through optimized angular light reception. This morphological engineering elevates interfacial energy flux density, directly intensifying evaporation kinetics. Building on principles analogous to those in RCYPSS, the corrugated architecture achieves hydraulic load optimization by maintaining a thin, uniform water film. This design concurrently mitigates parasitic thermal mass effects associated with bulk fluid retention, thereby altering the system's thermal time constant to prioritize latent heat transfer over sensible heat retention. The resultant thermodynamic optimization enhances vaporization kinetics while minimizing conductive losses to the aqueous phase. The geometric profile of corrugated cylinders in RCCYPSS reduces interstitial gas volume within the evaporation chamber by approximately 38% relative to PSS. This volumetric optimization decreases parasitic thermal inertia, enabling accelerated thermal response to incident irradiation and establishing a steeper vapor pressure gradient at the air-water interface. Experimental trials demonstrated a daily freshwater yield of 8500 mL/m² for RCCYPSS configurations, exceeding the 3100 mL/m² baseline of conventional PSS under identical insolation conditions – a 174% enhancement in productivity.

Critical analysis reveals that the corrugated morphology contributes a 30% incremental gain in distillate output when isolated from other system enhancements. This marginal improvement (174% total gain vs. 144% baseline performance without corrugation) underscores the synergistic role of surface geometry in amplifying interfacial thermodynamics while minimizing non-productive energy dissipation. The observed performance differential highlights the importance of chamber architecture in balancing radiative flux utilization with convective vapor transport efficiency.

### 3.3 Operation and assessment of rotating corrugated cylinders PSS (RCCPSS) with wick

When operating the system on the above experiments at the low drum speeds of 0.02 rpm (one loop every 50 minutes) and 0.05 rpm (one loop every 20 minutes), it was observed that the drum has dry spots formed on its outer surface due to the small motion. So, we tested the performance of the RCCPSS with a wick material covering the cylinders outer surface.

Fig 5 shows the total productivity improvement of the modified drum solar still with wick at 0.05 rpm drum speed. It is observed from Fig 5 that using the wick leads to improve the performance of the RCCPSS at the low drum speed. The increase in productivity rise records its value of 244% at 0.05 rpm. This improvement in distillate of MDSS with wick is

**Fig 5.** The total productivity improvement of the modified RCCPSS with wick.

higher than the highest value of productivity increase for RCCPSS without wick at 0.5 rpm (174%). This is maybe due to the slow motion at 0.05 rpm gives more opportunity for better evaporation without causing the dry spots phenomenon because of using the wick.

Tater production peaked at 14:00, with the RCPSS and PSS achieving maximum values of 1700 mL/m² & 710 mL/m², severally. Fig 4, further confirms that the total daily freshwater production of the RCPSS (10850 mL/m²) surpassed that of the PSS (3150 mL/m²). It means a 244% augment in distillate for RCCPSS with wick. This significant enhancement is primarily assigned to the superior vaporization rate accomplished by RCCPSS with wick compared to PSS.

### 3.4 Effect of raising the temperature of water of RCCPSS using external reflectors

Building upon the above observation that RCCPSS+ wick water temperature is consistently fewer than PSS by 0–7 °C, potentially hindering freshwater production, this study investigated the effectiveness of heating the basin water using external reflectors. To achieve maximum solar radiation concentration, the system was equipped with two external reflectors: one positioned on the front facade (external front reflector) and another on the backside (external back reflector).

The external reflectors in the RCCPSS configuration significantly elevated the basin water temperatures, consistently exceeding that of the PSS. The maximum temperatures differences between the water in the RCCPSS and PSS reached 6°C at 14:00. The highest temperature of the water occurred at 14:00, with values of 71 °C and 65 °C for the RCCPSS with reflectors and the PSS, respectively. Furthermore, the increased water temperature in the RCCPSS with reflectors led to a rise in vapor content generation by approximately 13 °C compared to the initial case without mirrors. Consequently, the glass temperatures of the RCPSS with mirrors also increased by 0–10 °C relative to the PSS, reaching a maximum of 58 °C at 14:00 compared to the PSS's 48 °C.

The improvement is assigned to the raised vapor generation within the RCCPSS due to the elevated water temperature facilitated by the external reflectors. As expected, peak hourly productivity occurred at 14:00, with the RCCPSS with reflectors achieving a maximum value of 1950 mL/m², exceeding the PSS's maximum of 760 mL/m².

The total distillate yields for the RCCPSS with wick and reflectors and the PSS were 13185 mL/m² and 3200 mL/m², respectively. It leads to 312% distillate augmentation for RCCPSS with reflectors. This significant enhancement is primarily

ascribed to the superior vaporization accomplished by RCCPSS due to the elevated water temperature facilitated by the reflectors. Notably, the inclusion of reflectors ensued 68% improvement in yield over the configuration without reflectors (312% with reflectors vs. 244% without).

### 3.5 Effect of raising the temperature of water of RCCPSS using heaters and PV system

We noticed from the previous results that the water temperature of the RCCPSS with wick was lower than that of the conventional pyramid solar still by 0–7 °C, and this indeed affects negatively the freshwater productivity. As a result, we wanted to investigate the influence of heating the basin water using three electric heaters with a power of 30 W each. The heaters were immersed into the basin water of the RCCPSS to heat the water of the still. Also, these heaters were run using a PV system with a power of 120 W. In addition, the minimum and maximum speeds of air ambient were 0.8 and 3.7 m/s, respectively, and the average air speed through the day was 2.1 m/s.

Fig 6 shows the distribution of solar radiation and temperatures of the PSS and RCCPSS, electric heaters, and PV system at 0.05 rpm. Comparing the results obtained from Fig 6 and Fig 7, incorporating the electric heaters into the basin water of the RCCPSS led to raise the basin water temperature, and it was always higher than that of the PSS. The readings revealed that the difference between the water temperatures of the RCCPSS and PSS with heaters reached 16 °C at 14:00 as illustrated in Fig 6. The maximum water temperature was obtained at 14:00, where it was 80 and 64 °C for the RCCPSS with heaters and PSS, respectively. Moreover, the raised water temperature of the RCCPSS with heaters led to increase the vapor content generation compared to the first case of RCCPSS without heaters. This increased the glass temperature over that of the PSS by 0–13 °C. The maximum glass temperature was obtained at 14:00, where it was 47 and 60 °C for the PSS and RCCPSS with heaters, respectively. Moreover, Fig 6 obtains the energy coming from the PV system to the heaters. It can be concluded from Fig 6 that the behavior of the PV energy is the same as that of the solar irradiance. Where it increases with increasing the solar radiation and decreases with the decline of solar intensity.

The hourly variation of the hourly and total freshwater productivity of the RSS and MPSSRC with electric heaters at cylinder speed of 0.05 rpm was illustrated in Fig 7. It can be observed from the figure that the hourly and total productivities of the MPSSRC with heaters were always greater than that of the PSS. This can be reasoned by the existence of the

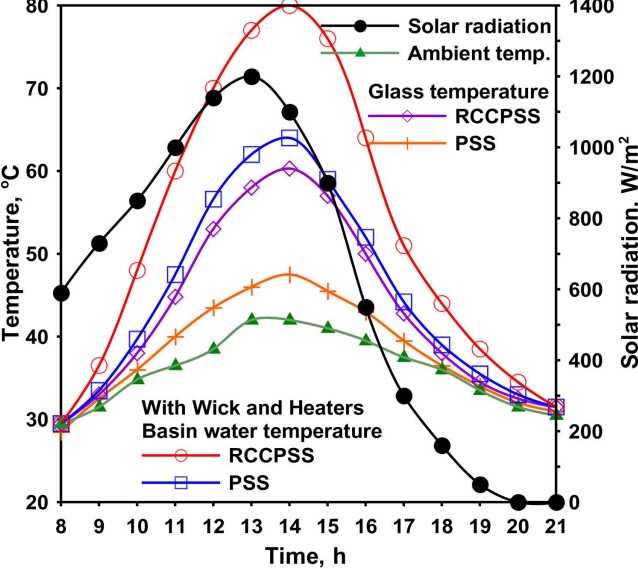

**Fig 6. Distribution of solar radiation and temperatures of solar stills with electric heaters and PV panel.**

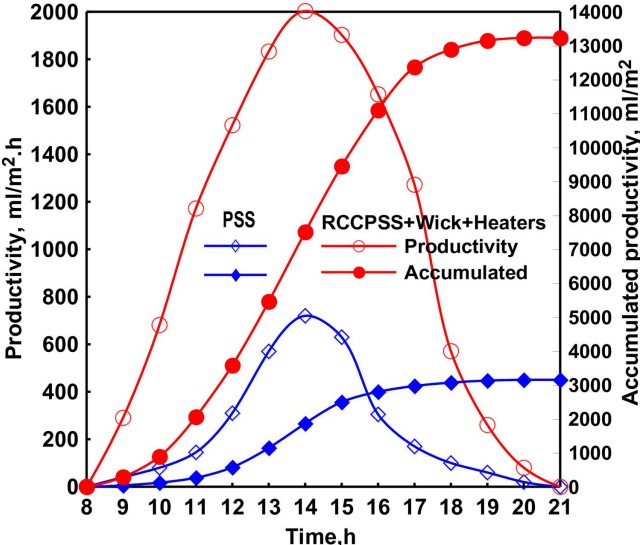

**Fig 7. Hourly and total productivity of solar stills with electric heaters and PV panel.**

electric heaters that augmented the vapor generation inside the solar still by raising the water temperature. As well, Fig 7 illustrates that the total productivity of the RCCPSS with electric heaters was more than that of the PSS. It was concluded that the total distillates of the RCCPSS with heaters and PSS were 13230 and 3150 mL/m² a day, respectively. So, the productivity was enhanced by 320%. This augmentation of productivity can be referred to the superior evaporation of the RCCPSS with heaters over that of the PSS.

This significant enhancement is primarily ascribed to the superior vaporization accomplished by RCCPSS with wick due to the elevated water temperature facilitated by the reflectors. Notably, the inclusion of electric heaters ensued 76% improvement in yield over the configuration without reflectors (320% with electric heaters vs. 244% without).

### 3.6 Operation of RCCPSS with PV panel and exhaust fan

Our investigation revealed that the RCCPSS design with external reflectors or heaters, while effective in raising basin water temperature, also led to a significant increase in glazing temperatures. The unexpectedly elevated temperature could hinder the RCPSS condensing procedure. To overcome it, an extra condensation device was implemented. This integrated system declined RCCPSS glazing heat. RCCPSS equipped with extra condensation had marginally higher glass cover over PSS by 0–13°C. Notably, they are substantially fewer than RCCPSS without extra condensation (increases of 0–6 °C, over PSS).

Two main processes are responsible for condenser's efficacy. Initially the removal fan lowers pressure inside RCPSS, which in turn causes the boiling to drop and, as a result, the amount of evaporating. Additionally, part of vapor produced within RCPSS is effectively removed by fan. Condensation on glazing is much reduced as a result of the lower vapor temperatures and decreased vapor level, which results in a general colder glassy temperature that is ideal for effective precipitation.

The results disclosed a remarkable yield improvement of RCPSS equipped with the external condenser and fan (RCPSS-fan) over PSS. The yield was augmented by 382% for RCPSS-fan, accomplishing a distillate of 14950 compared to only 3100 mL/m²·day for the pyramid SS. Notably, this represents a further improvement of roughly 62% in distillate referable solely to using extra condensation in RCPSS-fan.

## 3.6 Operation of RCCPSS with PV panel and paraffin wax-Ag

This study explores the potential of a composite material for thermal regulation within a hybrid cascade solar still (RCPSS). The composite material functions by absorbing excess thermal energy through high irradiation and vice versa. Fig 8 depicts the temperature fluctuations observed for RCPSS. These measurements include water, liner, & PCM. Different phases of heating and draining are shown by the obtained movements:

• Heating Stage (prior to 1:00 PM): The equipment as a whole experiences a continuous rise in temperature as sun energy increases in the direction of midday. The PCM aggressively collects heat from the liner over this heating period, gradually raising the internal temperature. PCM stores the heat that has been received as sensible heat.

• Releasing Stage (subsequent 1:00 PM): The PCM's temperature approaches its highest point and starts to decline, displaying three different stages as shown in Fig 8, following the peak in solar irradiation about 1:00 PM. When PCM approaches its hardening point, the temperature keeps dropping.

The heat that had been kept in PCM is released again at evening, raising the temperature of RCCPSS liquid. The drop in PCM's temperature is proof of this heat exchange. Notably, PCM temperature rises above the liners' towards 1:00 PM, emphasizing its function in storing heat and releasing it later on in the draining process. Through the PCM's method of thermal rules, RCCPSS's ultimate distillation effectiveness may be improved by maintaining a more steady temperature spectrum.

The incorporation of PCM with Nano alongside PV panel in the RCCPSS design resulted in a significant enhancement in fresh water amount over PSS. As showed in Fig 8, the daily fresh water yield for RCCPSS with Nano-PCM reached nearly 13950 mL/m².d. This represents 365% improvement in distillate over that of PSS (3000 mL/m².d). Notably, an estimated 45% increase in fresh water productivity results from this modification for the RCCPSS with PCM-Nano compared to the RCCPSS design without Nano-PCM Fig 9.

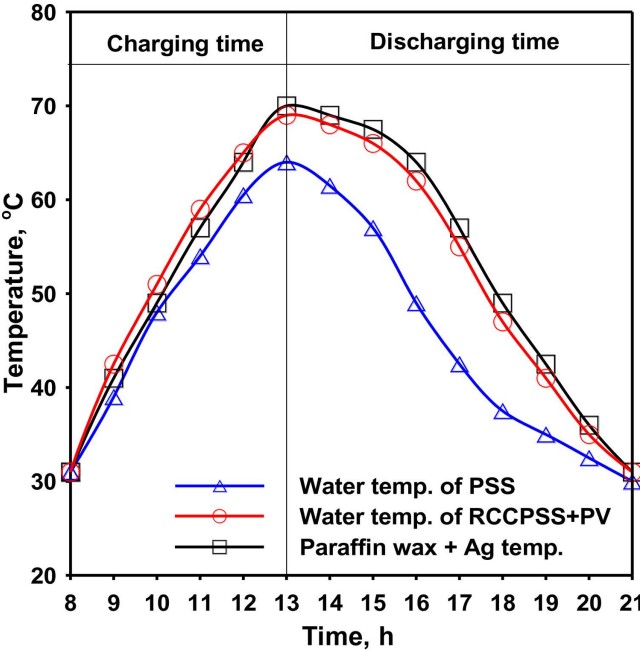

**Fig 8. Temperatures within PSS and RCCPSS with PV and PCM-Ag.**

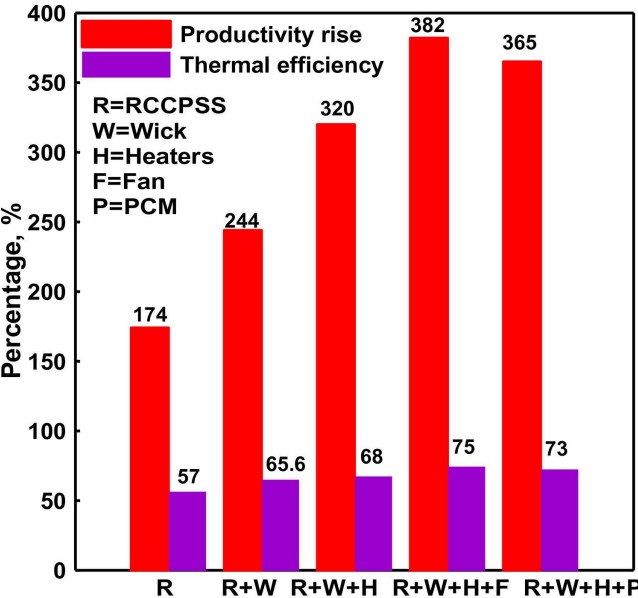

**Fig 9. Efficacy and distillate improvement of RCCPSS over PSS.**

### 3.5 Daily yield rise and Energy efficiency

According to reference [103], solar still performance is assessed through two key metrics: production increase and efficiency. The yield improvement is found as [104]:

$$Dailyrise, \% = \frac{RCCPSS\ yield - PSS\ yield}{PSS\ yield} \times 100$$

Also, the efficiency, ($\eta_{th}$), is found by [105,106].

$$\eta_{th} = \frac{\sum \dot{m} \times h_{fg}}{\sum (A \times I_R) + PV\ power}$$

Fig 6 depicts the improvements in efficiency and yield observed during the tests on the distillers. The thermal efficiency of the RCCPSS configurations increased progressively: 57% for the base design (RCCPSS), 65.6% for RCCPSS with wick, 68% for RCCPSS with heaters, 75% for RCPSS with heaters and a fan, and 73% for RCCPSS with the addition of PCM to the heaters and fan combination. Similarly, production increase followed the same trend, with values of 174%, 244%, 320%, 382% and 365% for the respective configurations. These results indicate that the combination of heaters and a fan yielded the most optimal performance for the RCCPSS system.

### 3.6 Cost analysis of distillate

Cost calculations were systematically performed to assess the economic practicality and viability of the RCCPSS system, utilizing comprehensive data presented in Table 3 in conjunction with the economic evaluation equations detailed in Table 4. The economic analysis reveals that the cost of freshwater production varies significantly between different system configurations. Specifically, the PSS demonstrates a water production cost of $0.02/L, while the RCCPSS equipped with a fan

**Table 3. Constant expense of distillers' components.**

| Distiller component | PSS ($) | RCPSS ($) |
|---|---|---|
| Iron | 20 | 20 |
| Glass | 15 | 15 |
| Fittings | 25 | 35 |
| Painting | 5 | 5 |
| Insulation | 10 | 15 |
| Cylinders | – | 40 |
| Production | 35 | 50 |
| Reflectors | – | 20 |
| Ag-Nanoparticles | – | 40 |
| Wax | – | 10 |
| Fan | – | 10 |
| DC motor | – | 10 |
| PV system | – | 50 |
| Heaters | – | 10 |
| Cooling copper coil | – | 10 |
| Total fixed cost (F) | **110** | **320** ($) for RCPSS + Reflector + PCM<br>**290** ($) for RCPSS + Reflector + Fan |

**Table 4. Analysis used in expenses analyses.**

| Variable | Meaning | Value |
|---|---|---|
| $n$ | Lifespan | 20, year |
| $i$ | Interest rate | 15% |
| $N$ | Days of operation | 340 d/year |
| $M$ | Annual production of water | 6050 L/m² for RCPSS + Reflector + PCM |
| | | 6300 L/m² for RCPSS + Reflector + Fan |
| | | 1300 L/m² for PSS |
| Freshwater cost (CPL) | | 0.0115 $/L for RCPSS + Reflector + PCM |
| | | 0.01 $/L for RCPSS + Reflector + Fan |
| | | 0.02 $/L for PSS |

achieves a notably lower production cost of $0.01/L. This 40% reduction in production cost for the RCCPSS with fan configuration can be attributed to the enhanced freshwater yield achieved through improved system efficiency, which effectively distributes the fixed capital costs over a larger volume of produced water. The economic advantage of the RCCPSS system, despite its higher initial capital investment due to additional components such as the fan and reflector system is clearly demonstrated through its superior cost-effectiveness in terms of per-liter water production costs. These results indicate that while the RCCPSS system requires higher upfront investment, the improved thermal performance and increased distillate yield result in better economic returns over the system's operational lifetime, making it a more economically viable solution for freshwater production applications.

### 3.7 Water quality analysis before and after distillation

The pH and total dissolved solids (TDS) of the feed water were analyzed before and after the desalination process. The results showed a decrease in pH from 9.5 (pre-distillation) to 7.4 (post-distillation), indicating a shift from alkaline

to near-neutral conditions. Similarly, the TDS levels dropped significantly from 1265 mg/L (pre-distillation) to 87 mg/L (post-distillation), demonstrating effective salt removal. These findings confirm that the distilled water meets the World Health Organization (WHO) guidelines [107] for acceptable drinking water quality.

## 4. Conclusions

This work examined the effect of different adjustments on the performance of pyramid distiller equipped with rotating corrugated cylinders (RCCPSS). The investigation included the effects of electrical heaters, PCM-Ag within the rotating cylinders, and a vapor withdrawing fan with an external condenser on the RCPSS's performance. The most significant remarks are as follows.

1. The total daily freshwater production of the RCCPSS (8500 mL/m²) surpassed that of PSS (3100 mL/m²). This means 174% yield improvement for RCCPSS.

2. The total daily freshwater production of the RCCPSS with wick (10850 mL/m²) surpassed that of PSS (3150 mL/m²). This means 244% yield improvement for RCCPSS.

3. The total daily freshwater production of the RCCPSS with wick and external reflectors (13185 mL/m²) surpassed that of PSS (3200 mL/m²). This means 312% yield improvement for RCCPSS.

4. The total distillate yields for the RCCPSS with wick and electrical heaters and the PSS were 13230 mL/m² and 3150 mL/m², respectively. This translates to a remarkable 320% increase in production for the RCCPSS with heaters. Notably, the inclusion of heaters resulted in a 76% improvement in production compared to the configuration without reflectors (320% with reflectors vs. 244% without).

5. The average daily freshwater yield for the RCCPSS with Nano-PCM reached nearly 13950 mL/m²·d. This represents a remarkable 365% increase in production compared to the yield observed for the standalone PSS (3000 mL/m²·d). Notably, this improvement translates to an estimated 45% rise in fresh water production for the RCCPSS in contrast to the RCCPSS design without Nano-PCM, using PCM-Nano.

6. The combination of heaters and a fan yielded the most optimal performance for the RCPSS system, where the distillate improvement and efficacy were 382% and 75%, respectively.

7. The expenses of water are 0.02 and 0.01$/L for the PSS and the RCCPSS with wick, heaters and fan, severally.

While this study demonstrates significant improvements in pyramid solar still (PSS) performance through rotating cylinders and advanced modifications, several limitations should be acknowledged:

1. Mechanical complexity of rotating cylinders

   • The manufacturing and installation of rotating cylinders—whether flat or core-jetted—pose significant engineering challenges due to their dynamic nature.

   • Rotation introduces a risk of fluid leakage, requiring high-precision sealing mechanisms to maintain system integrity.

2. System complexity from integrated components

   • The system's design involves intricate internal components (e.g., rotating cylinders) and external elements (e.g., mirrors and condensers), increasing assembly and maintenance difficulties.

   • Coordination between moving and stationary parts demands robust alignment and synchronization to ensure optimal performance.

  

3. Nanomaterial leakage risks in paraffin wax

- When nanomaterials are dispersed in paraffin wax, there is a potential for leakage or phase separation, particularly under thermal or mechanical stress.

- This risk necessitates advanced encapsulation techniques or barrier designs to prevent contamination or efficiency losses.

Future work will explore the performance of puffin and half-barrel cylinders inside the pyramid solar still, with detailed analysis to be included in subsequent research. In addition, other commercial types of PCM can be investigated.

## Acknowledgments

The authors extend their appreciation to Prince Sattam bin Abdulaziz University for funding this research work through the project number (PSAU/2025/01/34686).

## Author contributions

**Investigation:** A.S. Abdullah, Mutabe Aljaghtham, Z. M. Omara, Fadl A. Essa.

**Methodology:** A.S. Abdullah, Mutabe Aljaghtham.

**Project administration:** Z. M. Omara.

**Resources:** A.S. Abdullah, Mutabe Aljaghtham, Fadl A. Essa, Wissam H. Alawee.

**Supervision:** A.S. Abdullah, Wissam H. Alawee.

**Writing – original draft:** A.S. Abdullah, Fadl A. Essa, Wissam H. Alawee.

**Writing – review & editing:** Mutabe Aljaghtham, Fadl A. Essa.

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
