## [Decision Letter · Decision Letter 0]

23 Jun 2025

PONE-D-25-25343Optimizing pyramid distiller performance with rotational corrugated cylinders, reflectors, and PCM-fan integration for enhanced freshwater productionPLOS ONE

Dear Dr. Omara,

Thank you for submitting your manuscript to PLOS ONE. After careful consideration, we feel that it has merit but does not fully meet PLOS ONE’s publication criteria as it currently stands. Therefore, we invite you to submit a revised version of the manuscript that addresses the points raised during the review process.

We look forward to receiving your revised manuscript.

Kind regards,

S. Shanmugan, PhD

Academic Editor

PLOS ONE

Additional Editor Comments:

Dear Z. M. Omara (Author)

Reviewer #1

1) The authors should follow the journal guidelines

2) Highlights are missing: each bullet should be 85 characters with spaces.

3) Avoid abbreviation in the title and abstract (if possible).

4) A comprehensive proofreading must be conducted for the text of the manuscript by a technical native person.

5) The introduction section warrants a more comprehensive treatment, incorporating recent literature to enhance its contextualization and relevance.

6) In Literature, The author should brought many irrelevant external modifications as PCM, reflectors or nanomaterials are not considered as the design parameters rather all of them are external modification. The authors should include recent studies such as

- Performance assessment of a novel solar distiller with a fountain-shaped basin design embedded with phase change materials enriched with copper oxide nano-additives: A detailed experimental investigation, Journal of Energy Storage, 2024

- Predicting the yield of stepped corrugated solar distiller using kernel-based machine learning models, Applied Thermal Engineering, 2023.

- Optimal size of spherical rock salt balls as low-cost thermal storage materials for performance augmentation of hemispherical solar distillers: Experimental investigation and thermo-economic analysis, Journal of Cleaner Production, 2022.

- Improving the thermo-economic performance of hemispherical solar distiller using copper oxide nanofluids and phase change materials: Experimental and theoretical investigation, Solar Energy Materials and Solar Cells, 2022.

7) The distinctive aspects of novelty in the current study should be emphasized more prominently.

8) It is imperative to incorporate a detailed Table regarding the distiller geometrical details and properties of the utilized thermal energy storage material.

9) Elaborate on the performance degradation observed in the modifications of the study for a comprehensive analysis of patterns.

10) Strengthen the discussion section by delving into the mechanisms underlying the observed phenomena, thereby elevating the overall quality of the manuscript.

11) Extend the discussion to encompass water quality analyses of both seawater and distillate output, offering a more comprehensive perspective on the study's broader implications.

12) It is advisable to include the limitations of the study.

Reviewer #2

Paper Title (Optimizing pyramid distiller performance with rotational corrugated cylinders, reflectors, and pcm-fan integration for enhanced freshwater production). The topic and novelty of the work is interesting and the topic aligns with the scope of Pols One Journal. I have some comments which I want the authors to work on before publishing it in the journal.

1- The title should be more descriptive.

2- The originality of this paper should be clarified in the introduction section.

3- Error analysis is mandatory for any experimental work. Please include.

4- The references do not follow the journal style.

5- Check the subscripts and superscripts through the entire context.

6- The manuscript's English has to be strengthened. The content is riddled with typos and syntactical flaws.

7- The rotating cylinders must by studied with wick materials

8- Used others type of cylinder surface such as half- barrel

9- What happens if the exterior mirrors are replaced with PV-cells to heat the water for the still?

10- Quantity of PCM material (in kgs) used in the Solar stills not specified. Also, properties of PCM and Ag-nano particles not specified. In addition, properties of PCM + Ag-nano particles together not specified.

Reviewers' comments:

Reviewer's Responses to Questions

**Comments to the Author**

1. Is the manuscript technically sound, and do the data support the conclusions?

Reviewer #1: Partly

Reviewer #2: Yes

2. Has the statistical analysis been performed appropriately and rigorously?

Reviewer #1: Yes

Reviewer #2: N/A

3. Have the authors made all data underlying the findings in their manuscript fully available?

Reviewer #1: Yes

Reviewer #2: Yes

4. Is the manuscript presented in an intelligible fashion and written in standard English?

Reviewer #1: No

Reviewer #2: Yes

5. Review Comments to the Author

Reviewer #1: 1) The authors should follow the journal guidelines

2) Highlights are missing: each bullet should be 85 characters with spaces.

3) Avoid abbreviation in the title and abstract (if possible).

4) A comprehensive proofreading must be conducted for the text of the manuscript by a technical native person.

5) The introduction section warrants a more comprehensive treatment, incorporating recent literature to enhance its contextualization and relevance.

6) In Literature, The author should brought many irrelevant external modifications as PCM, reflectors or nanomaterials are not considered as the design parameters rather all of them are external modification. The authors should include recent studies such as

- Performance assessment of a novel solar distiller with a fountain-shaped basin design embedded with phase change materials enriched with copper oxide nano-additives: A detailed experimental investigation, Journal of Energy Storage, 2024

- Predicting the yield of stepped corrugated solar distiller using kernel-based machine learning models, Applied Thermal Engineering, 2023.

- Optimal size of spherical rock salt balls as low-cost thermal storage materials for performance augmentation of hemispherical solar distillers: Experimental investigation and thermo-economic analysis, Journal of Cleaner Production, 2022.

- Improving the thermo-economic performance of hemispherical solar distiller using copper oxide nanofluids and phase change materials: Experimental and theoretical investigation, Solar Energy Materials and Solar Cells, 2022.

7) The distinctive aspects of novelty in the current study should be emphasized more prominently.

8) It is imperative to incorporate a detailed Table regarding the distiller geometrical details and properties of the utilized thermal energy storage material.

9) Elaborate on the performance degradation observed in the modifications of the study for a comprehensive analysis of patterns.

10) Strengthen the discussion section by delving into the mechanisms underlying the observed phenomena, thereby elevating the overall quality of the manuscript.

11) Extend the discussion to encompass water quality analyses of both seawater and distillate output, offering a more comprehensive perspective on the study's broader implications.

12) It is advisable to include the limitations of the study.

Reviewer #2: Paper Title (Optimizing pyramid distiller performance with rotational corrugated cylinders, reflectors, and pcm-fan integration for enhanced freshwater production). The topic and novelty of the work is interesting and the topic aligns with the scope of Pols One Journal. I have some comments which I want the authors to work on before publishing it in the journal.

1- The title should be more descriptive.

2- The originality of this paper should be clarified in the introduction section.

3- Error analysis is mandatory for any experimental work. Please include.

4- The references do not follow the journal style.

5- Check the subscripts and superscripts through the entire context.

6- The manuscript's English has to be strengthened. The content is riddled with typos and syntactical flaws.

7- The rotating cylinders must by studied with wick materials

8- Used others type of cylinder surface such as half- barrel

9- What happens if the exterior mirrors are replaced with PV-cells to heat the water for the still?

10- Quantity of PCM material (in kgs) used in the Solar stills not specified. Also, properties of PCM and Ag-nano particles not specified. In addition, properties of PCM + Ag-nano particles together not specified.

6. PLOS authors have the option to publish the peer review history of their article (what does this mean?). If published, this will include your full peer review and any attached files.

Reviewer #1: No

Reviewer #2: No

---

## [Author Response · Author response to Decision Letter 1]

9 Jul 2025

Responses to Reviewers’ comments

PONE-D-25-25343

Optimizing pyramid distiller performance with rotational corrugated cylinders, reflectors, and PCM-fan integration for enhanced freshwater production

Dear Editor of PLOS One Journal.

Wissam H. Alawee: was work hard on revised manuscript, so we hope that the Editor will approve of adding his to this work.

Reviewer #1

Q1) The authors should follow the journal guidelines

A1: Thanks for the comment. The journal guidelines are followed as required by the respected reviewer.

Q2) Highlights are missing: each bullet should be 85 characters with spaces.

A2: Thanks for the comment. The highlights are revised to be as suggested. Kindly refer to the revised Highlights.

Q3) Avoid abbreviation in the title and abstract (if possible).

A3: Thanks for the comment. The abbreviations are avoided as required.

Q4) A comprehensive proofreading must be conducted for the text of the manuscript by a technical native person.

A4: Thanks for the comment. The manuscript is well revised to polish the language and avoid mistakes and errors as possible.

Q5) The introduction section wants a more comprehensive treatment, incorporating recent literature to enhance its contextualization and relevance.

A5: Thanks for the comment. The introduction section is revised and edited as required by the respected reviewer.

Q6) In Literature, the author should brought many irrelevant external modifications as PCM, reflectors or nanomaterials are not considered as the design parameters rather all of them are external modification. The authors should include recent studies such as

Performance assessment of a novel solar distiller with a fountain-shaped basin design embedded with phase change materials enriched with copper oxide nano-additives: A detailed experimental investigation, Journal of Energy Storage, 2024

Predicting the yield of stepped corrugated solar distiller using kernel-based machine learning models, Applied Thermal Engineering, 2023.

Optimal size of spherical rock salt balls as low-cost thermal storage materials for performance augmentation of hemispherical solar distillers: Experimental investigation and thermo-economic analysis, Journal of Cleaner Production, 2022.

Improving the thermo-economic performance of hemispherical solar distiller using copper oxide nanofluids and phase change materials: Experimental and theoretical investigation, Solar Energy Materials and Solar Cells, 2022.

A6: Thanks for the comment. The given recent studies are used to boost the literature.

Q7) The distinctive aspects of novelty in the current study should be emphasized more prominently.

A7: We sincerely appreciate the reviewer’s constructive feedback. In response to comment Q7, we have revised the manuscript to more prominently highlight the novel aspects of our study, particularly the innovative use of rotating cylinders with corrugated surfaces, the discovery of distinct optimal rotational speeds, and the integration of advanced modifications such as PCM-Ag nanocomposites and vapor extraction. These contributions represent significant departures from conventional PSS designs and have been explicitly emphasized in the revised text (see last Section of Introduction). We hope these clarifications better underscore the originality and impact of our work.

Q8) It is imperative to incorporate a detailed Table regarding the distiller geometrical details and properties of the utilized thermal energy storage material.

A8: Thanks for the valuable comment. These data are added in tables as required. Kindly refer to Tables 1 and 2 under section 2.1.

Q9) Elaborate on the performance degradation observed in the modifications of the study for a comprehensive analysis of patterns.

A9: We thank the reviewer for raising this important point. In response to Q9, we have expanded the discussion on performance degradation mechanisms to include a detailed analysis of PSS performance under the testing conditions. These insights provide a clearer understanding of the limitations and optimization pathways for the proposed modifications. Kindly refer to section 3.

Q10) Strengthen the discussion section by delving into the mechanisms underlying the observed phenomena, thereby elevating the overall quality of the manuscript.

A10: Thanks for the comment. The results and discussion section is strengthened as suggested. Kindly refer to section 3.

Q11) Extend the discussion to encompass water quality analyses of both seawater and distillate output, offering a more comprehensive perspective on the study's broader implications.

A11: Thanks for the comment. The manuscript is edited to have it under a section 3.7.

3.7 Water quality analysis before and after distillation

The pH and total dissolved solids (TDS) of the feed water were analyzed before and after the desalination process. The results showed a decrease in pH from 9.5 (pre-distillation) to 7.4 (post-distillation), indicating a shift from alkaline to near-neutral conditions. Similarly, the TDS levels dropped significantly from 1265 mg/L (pre-distillation) to 87 mg/L (post-distillation), demonstrating effective salt removal. These findings confirm that the distilled water meets the World Health Organization (WHO) guidelines [104] for acceptable drinking water quality.

Q12) It is advisable to include the limitations of the study.

A12: We appreciate the reviewer’s suggestion to elaborate on limitations. A dedicated paragraph at the end of the conclusion section now addresses, aligning with the journal’s emphasis on practical applicability. We believe this addition provides a more balanced perspective.

Reviewer #2

Q1- The title should be more descriptive.

A1: Thanks for the comment. The title is edited to be more descriptive as required.

Q2- The originality of this paper should be clarified in the introduction section.

A2: We sincerely appreciate the reviewer’s constructive feedback. In response to comment Q2, we have revised the manuscript to more prominently highlight the novel aspects of our study, particularly the innovative use of rotating cylinders with corrugated surfaces, the discovery of distinct optimal rotational speeds, and the integration of advanced modifications such as PCM-Ag nanocomposites and vapor extraction. These contributions represent significant departures from conventional PSS designs and have been explicitly emphasized in the revised text (see last Section of Introduction). We hope these clarifications better underscore the originality and impact of our work.

Q3- Error analysis is mandatory for any experimental work. Please include.

A3: Thanks for the comment. The error analysis is included as required in the revised version of the manuscript. Kindly refer to section 2.3.

Q4- The references do not follow the journal style.

A4: Thanks for the comment. The reference style is edited to follow the journal style.

Q5- Check the subscripts and superscripts through the entire context.

A5: Thanks for the comment. The subscripts and superscripts are revised through the entire context of the manuscript.

Q6- The manuscript's English has to be strengthened. The content is riddled with typos and syntactical flaws.

A6: The language is well revised to be strengthened as required.

Q7- The rotating cylinders must be studied with wick materials

A7: We sincerely appreciate the reviewer’s constructive feedback. In response to comment Q7, we have studied the rotating cylinders with wick materials. Kindly refer to section 3.3.

Q8- Used others type of cylinder surface such as half- barrel.

A8: We sincerely appreciate the reviewer’s constructive feedback. In this investigation, two-cylinder configurations were employed: flat and corrugated. The corrugated design was prioritized due to its manufacturing feasibility compared to more complex geometries. While a half-barrel cylinder could offer a larger heat transfer surface area, its fabrication presents significant challenges, making it less practical for the current scope. Future work will explore the performance of puffin and half-barrel cylinders, with detailed analysis to be included in subsequent research. Kindly refer to the last of the conclusions section (future work).

Q9- What happens if the exterior mirrors are replaced with PV-cells to heat the water for the still?

A9: Thanks for the comment. As the reviewer required, the effect of raising the temperature of water of RCCPSS using heaters and PV system in added in the revised manuscript. Kindly refer to section 3.5.

Q10- Quantity of PCM material (in kgs) used in the Solar stills not specified. Also, properties of PCM and Ag-nano particles not specified. In addition, properties of PCM + Ag-nano particles together not specified.

A10: Thanks for the comment. The quantity of PCM material used in solar stills is added as required. In addition, the properties of PCM with and without nanomaterials are added in the revised version. Kindly refer to section 2.1 and Table 2.

---

## [Decision Letter · Decision Letter 1]

27 Jul 2025

Experimental Investigation of Shape-Enhanced Rotating Cylinders with Electric Heaters and Solar Panels for Augmented Pyramid Solar Still Performance

PONE-D-25-25343R1

Dear Dr. Z. M. Omara,

We’re pleased to inform you that your manuscript has been judged scientifically suitable for publication and will be formally accepted for publication once it meets all outstanding technical requirements.

Kind regards,

S. Shanmugan, PhD

Academic Editor

PLOS ONE

Additional Editor Comments (optional):

Accept

Reviewers' comments:

Reviewer's Responses to Questions

**Comments to the Author**

1. If the authors have adequately addressed your comments raised in a previous round of review and you feel that this manuscript is now acceptable for publication, you may indicate that here to bypass the “Comments to the Author” section, enter your conflict of interest statement in the “Confidential to Editor” section, and submit your "Accept" recommendation.

Reviewer #1: All comments have been addressed

Reviewer #2: All comments have been addressed

2. Is the manuscript technically sound, and do the data support the conclusions?

Reviewer #1: Yes

Reviewer #2: Yes

3. Has the statistical analysis been performed appropriately and rigorously?

Reviewer #1: N/A

Reviewer #2: Yes

4. Have the authors made all data underlying the findings in their manuscript fully available?

Reviewer #1: Yes

Reviewer #2: Yes

5. Is the manuscript presented in an intelligible fashion and written in standard English?

Reviewer #1: Yes

Reviewer #2: (No Response)

6. Review Comments to the Author

Reviewer #1: No further comments are required. It can be accepted for publication in the journal

Reviewer #2: The researcher has responded to all questions satisfactorily, there are no further comments, and the research is accepted.

7. PLOS authors have the option to publish the peer review history of their article (what does this mean?). If published, this will include your full peer review and any attached files.

Reviewer #1: No

Reviewer #2: No

---

## [Editor Report · Acceptance letter]

PONE-D-25-25343R1

PLOS One

Dear Dr. Omara,

I'm pleased to inform you that your manuscript has been deemed suitable for publication in PLOS One. Congratulations! Your manuscript is now being handed over to our production team.

Kind regards,

on behalf of

Dr. S. Shanmugan

Academic Editor

PLOS One